# YTHDF2 as a Mediator in BDNF-Induced Proliferation of Porcine Follicular Granulosa Cells

**DOI:** 10.3390/ijms25042343

**Published:** 2024-02-16

**Authors:** Kening Liu, Xu Zhou, Chunjin Li, Caomeihui Shen, Guitian He, Tong Chen, Maosheng Cao, Xue Chen, Boqi Zhang, Lu Chen

**Affiliations:** College of Animal Science, Jilin University, Changchun 130062, China; liukn20@mails.jlu.edu.cn (K.L.); zxu@jlu.edu.cn (X.Z.); li_chunjin@jlu.edu.cn (C.L.); scmh21@mails.jlu.edu.cn (C.S.); hegt22@mails.jlu.edu.cn (G.H.); chentong21@mails.jlu.edu.cn (T.C.); caoms21@mails.jlu.edu.cn (M.C.); xuechen21@mails.jlu.edu.cn (X.C.); zzhangbq@jlu.edu.cn (B.Z.)

**Keywords:** granulosa cells, brain-derived neurotrophic factor (BDNF), follicular development

## Abstract

In female mammals, the proliferation and apoptosis of granulosa cells (GCs) are critical in determining the fate of follicles and are influenced by various factors, including brain-derived neurotrophic factor (BDNF). Previous research has shown that BDNF primarily regulates GC proliferation through the PI3K/AKT, NF-kB, and CREB tumour pathways; however, the role of other molecular mechanisms in mediating BDNF-induced GC proliferation remains unclear. In this study, we investigated the involvement of the m^6^A reader YTH domain-containing family member 2 (YTHDF2) in BDNF-stimulated GC proliferation and its underlying mechanism. GCs were cultured in DMEM medium supplemented with varying BDNF concentrations (0, 10, 30, 75, and 150 ng/mL) for 24 h. The viability, number, and cell cycle of GCs were assessed using the CCK-8 assay, cell counting, and flow cytometry, respectively. Further exploration into *YTHDF2’s* role in BDNF-stimulated GC proliferation was conducted using RT-qPCR, Western blotting, and sequencing. Our findings indicate that *YTHDF2* mediates the effect of BDNF on GC proliferation. Additionally, this study suggests for the first time that BDNF promotes *YTHDF2* expression by increasing the phosphorylation level of the ERK1/2 signalling pathway. This study offers a new perspective and foundation for further elucidating the mechanism by which BDNF regulates GC proliferation.

## 1. Introduction

Studies have demonstrated that brain-derived neurotrophic factor (BDNF) [1], nerve growth factor (NGF) [2], neurotrophin 3 (NT-3) [3,4], and neurotrophin 4/5 (NT-4/5) [5] play significant roles in the regulation of mammalian reproduction. Specifically in *pigs*, the BDNF gene is located on chromosome 6 and spans approximately 55 kb, comprising six exons. Pro-BDNF predominantly binds to the p75 receptor, while mature BDNF primarily associates with tyrosine kinase receptor protein B (TrkB) which is found in most mammals [6]. An early study confirmed BDNF’s presence in *porcine* ovarian granulosa cells and oocytes [7]. It is established that glucocorticoids and oestrogen are central regulators of BDNF. Follicle-stimulating hormone (FSH) and human chorionic gonadotropin (hCG) can enhance BDNF synthesis from granulosa cells (GCs) in *humans* [8]. BDNF facilitates the proliferation of *bovine* follicular GCs and progesterone production [9]. In *Bacillus bubalis*, BDNF promotes oocyte maturation and early embryonic development [10]. Therefore, BDNF is integral to GC cell proliferation and oocyte maturation. Previous research has revealed that BDNF regulates GC proliferation through several signalling pathways (PI3K/AKT, NF-kB, and CREB), and that cytokines like PTEN and GDF-9 can mediate BDNF’s action [1]; however, the role of BDNF in regulating GC proliferation through m^6^A methylase, demethylase, and readers—key regulatory factors for mammalian reproduction—remains unreported [11].

m^6^A methylases, including METTL3, METTL14, METTL16, KIAA1429, and WTAP, are responsible for increasing m^6^A methylation modifications [12]. In *pigs*, METTL3-mediated m^6^A methylation regulates autophagy in GCs during follicular atresia [13]. In *mice*, METTL3-mediated m^6^A modification is essential for oocyte and follicle development [14]. m^6^A demethylases such as ALKBH5 and FTO decrease m^6^A methylation modifications [15], with FTO implicated in ovarian granulosa cell dysfunction via upregulating flotillin 2 [16]. m^6^A readers include YTH domain-containing family members 1, 2, and 3 (YTHDF1, YTHDF2, andYTHDF3), and YTH domain-containing 1 and 2 (YTHDC1 and YTHDC2), all of which are involved in the m^6^A methylation processes [17]. In *mice*, YTHDC1-deficient oocytes are hindered at the primary follicular stage due to numerous alternative splicing deficiencies [18], and adult *female mice* lacking YTHDC2 are infertile [19]. YTHDC2 influences the expression of meiotic markers and the proportion of female germ cells at the zygotene stage [19]. YTHDF2 upregulates FOXO3 mRNA in GCs and contributes to the progression of non-obese polycystic ovary syndrome (PCOS) in *women* [20]. Moreover, YTHDF2 plays a vital role in oocyte maturation and early fertilization during egg development in *mice* [21], and it is involved in various biological processes such as proliferation, apoptosis, and differentiation [22].

This study investigates the role of the m^6^A reader YTHDF2 in BDNF-stimulated GC proliferation and the mechanism by which BDNF regulates YTHDF2 expression. To our knowledge, this is the first study demonstrating that BDNF upregulates YTHDF2 expression via the extracellular regulated protein kinases 1/2 (ERK1/2) pathway. Additionally, YTHDF2 enhances Cyclin-dependent Kinase 4 (CDK4) and Proliferating Cell Nuclear Antigen (PCNA) expression, thus promoting GC proliferation. This provides a novel perspective and foundation for further understanding the role of BDNF in the regulation of follicular GC proliferation.

## 2. Results

### 2.1. BDNF Promoted the Proliferation and Upregulated the Expression of YTHDF2 in GCs

A specific GC protein, FSHR, was employed as a marker for the characterization of isolated GCs. Fluorescence microscopy results revealed that the anti-FSHR treated group exhibited strong green fluorescence compared to the negative control group (Figure 1A). CCK-8 and cell-counting assays were conducted to assess the impact of BDNF on the viability and number of GCs. Compared to the control group, the BDNF treatment at concentrations of 75 ng/mL and 150 ng/mL enhanced the viability and number of GCs (Figure 1B,C); however, the 75 ng/mL BDNF group displayed a higher cell viability and number than the 150 ng/mL BDNF group. Consequently, 75 ng/mL BDNF was selected for subsequent experiments. Flow cytometry analysis revealed that the percentage of cells in the G1 phase significantly decreased, while the percentage in the S phase increased markedly, and the G2 phase showed a slight but non-significant increase in BDNF-treated GCs (Figure 1D). These findings suggest that BDNF enhances the viability and proliferation of GCs. In this study, we systematically screened the expression levels of m^6^A methylases, demethylases, and readers following BDNF treatment. The results demonstrated that the methylation recognition protein YTHDF2 was significantly upregulated in the BDNF-treated group. In contrast, there were no significant changes in the mRNA levels of the methylases METTL3 and METTL14; demethylases FTO and ALKBH5; and readers YTHDC1, YTHDC2, YTHDF1, and YTHDF3 (Figure 1E).

### 2.2. YTHDF2 Knockdown Impacts Proliferation and Gene Expression in GCs

A *YTHDF2 knockdown* (siYTHDF2) model in GCs was established to investigate the regulatory effects of YTHDF2 on GC proliferation. This model was validated using RT-qPCR and Western blotting (Figure 2A,B). In the siYTHDF2 group, GC viability, cell number, and cell cycle progression were significantly reduced compared to their progression in the control group (Figure 2C–E). Transcriptomic sequencing revealed that in the siYTHDF2 group, 322 genes were significantly downregulated, and 104 genes were significantly upregulated relative to the control group (Figure 3A). The heatmap illustrates the differential gene expression between the control and siYTHDF2 groups (Figure 3B). A KEGG enrichment analysis identified the PI3K-Akt signalling pathway as being significantly enriched (Figure 3C). A Gene Ontology (GO) enrichment analysis highlighted significant enrichment in the cell proliferation and apoptosis categories (Figure 3D). These findings suggest that YTHDF2 plays a crucial role in the viability and proliferation of GCs.

### 2.3. YTHDF2 Facilitates GC Proliferation by Upregulating CDK4 and PCNA

The role of YTHDF2 in regulating GC proliferation was further investigated. YTHDF2 and the top six differentially expressed genes (*TMEM255A, CLDN11, FN1, NOX4, PCNA, and CDK4*) within the PI3K-Akt signalling pathway were validated through qPCR. These findings aligned with our sequencing data (Figure 4A). Notably, the mRNA and protein levels of cell-cycle-dependent kinase 4 (*CDK4*) and Proliferating Cell Nuclear Antigen (*PCNA*), both of which vital for cell cycle regulation, were significantly reduced in the *YTHDF2* knockdown group compared to the control group (Figure 4A,B). In contrast, the mRNA and protein levels of *CDK4 and PCNA* were significantly increased in the BDNF treatment group relative to the control group (Figure 4C,D). Additional results revealed that cell viability, S phase entry, and cell count were markedly inhibited in the siYTHDF2 group versus the control group; however, there were no significant differences between the siYTHDF2 + BDNF group and the control group in these aspects (Figure 4E–G). Similarly, the mRNA and protein levels of *PCNA and CDK4* showed no significant differences between the siYTHDF2 + BDNF and control groups (Figure 4H,I). These outcomes suggest that YTHDF2 plays a crucial role in mediating BDNF-induced GC proliferation by upregulating *CDK4 and PCNA*.

### 2.4. BDNF Regulates YTHDF2 through the ERK1/2 Signalling Pathway

To delve deeper into the molecular mechanism by which BDNF regulates *YTHDF2*, we examined the expression of total ERK1/2 (t-ERK1/2) and phosphorylated ERK1/2 (P-ERK1/2) in GCs following BDNF treatment. Our results indicated a significantly higher (P-ERK1/2)/(t-ERK1/2) ratio in the BDNF-treated group compared to the control group; however, the (t-ERK1/2)/(vinculin) ratio did not show a significant difference between the BDNF and control groups (Figure 5A). We also observed that the expression of both t-ERK1/2 and P-ERK1/2 was notably inhibited in the PD98059 group (Figure 5B). Moreover, the mRNA and protein levels of *YTHDF2* were substantially reduced in the PD98059 group compared with the control group (Figure 5B). These findings led us to speculate that BDNF mediates the regulation of *YTHDF2* via the ERK1/2 signalling pathway. As anticipated, further analysis revealed a significant increase in cell viability and the proportion of cells in the S phase in the BDNF group, while these phenotypes were reversed in the BDNF + PD 98059 group (Figure 5C,D). The mRNA and protein expression levels of YTHDF2 were significantly elevated in the BDNF group relative to the control group and were markedly reduced in the BDNF + PD 98059 group compared to the BDNF group. Importantly, there was no significant difference between the BDNF+PD98059 group and the control groups (Figure 5E,F). Further investigation confirmed that the mRNA and protein expression levels of PCNA and CDK4 were significantly higher in the BDNF group than in the control group and were significantly decreased in the BDNF + PD 98059 group compared to the BDNF group (Figure 5E,F). These results collectively suggest that BDNF modulates YTHDF2 expression through the activation of the ERK1/2 signalling pathway.

## 3. Discussion

To our knowledge, this is the first study to assess the role of *YTHDF2* in mediating the effects of BDNF on the proliferation of *porcine* follicular granulosa cells. This study demonstrates that *YTHDF2* is crucial in BDNF-induced GC proliferation, as evidenced by its impact on cell viability, number, and cycle. These findings align with prior studies confirming BDNF’s presence in GCs and oocytes of *porcine* ovaries [7], typically associated with GC proliferation and oocyte maturation [23]. BDNF has been shown to promote GC proliferation in *cattle* and *goats* [9,24] and Leydig cell proliferation in *mice* [25]. Furthermore, BDNF appears to enhance the proliferation of various cells within the reproductive system. In this study, treating GCs with 75 ng/mL and 150 ng/mL BDNF for 24 h significantly stimulated their proliferation, reinforcing this conclusion.

m^6^A modification is critical for oogenesis, as indicated by the marked enrichment of differentially expressed m^6^A methylation genes in pathways related to granulosa cell proliferation and follicular development [26]. YTH domain proteins and their family members play a significant role in recognizing m^6^A modification sites and regulating the reproductive system. In *mice*, the absence of *YTHDF2*, essential for producing MⅡ oocytes capable of sustaining early zygotic development, results in the failure of mRNA degradation in some m^6^A-modified oocytes, adversely affecting oocyte quality [21]. Our study suggests that BDNF, via *YTHDF2* upregulation, promotes GC proliferation, and that *YTHDF2* downregulation significantly hinders cell cycle progression. This finding is consistent with previous research showing that *YTHDF2* facilitates mitotic entry by regulating the cell cycle [27]. In multiple myeloma cells, *YTHDF2* sustains proliferation through the STAT5A/MAP2K2/p-ERK pathway [28], while in lung adenocarcinoma cells, it promotes proliferation by targeting the AXIN1/WNT/β-catenin signalling pathway. Conversely, other studies have shown that *YTHDF2* inhibits cell proliferation and growth in hepatocellular carcinoma by destabilizing EGFR mRNA [29]. Thus, current perspectives on *YTHDF2’s* role in cell cycle regulation are mixed, and our findings help clarify this mechanism. These results also imply that BDNF may regulate GC proliferation through m^6^A methylases, demethylases, and readers, warranting further investigation.

Our results suggest that *YTHDF2* enhances the BDNF-induced viability and proliferation of GCs by upregulating *CDK4 and PCNA* expression. Flow cytometry analysis revealed that increased YTHDF2 expression promotes the transition of cells from the G1 to the S phase, thereby enhancing GC viability. Conversely, this transition was impeded when *YTHDF2* was knocked down. Thus, it appears that *YTHDF2* regulates the G1-to-S phase transition in GCs through the upregulation of *CDK4 and PCNA*. Regarding cell cycle regulation, studies have demonstrated that *YTHDF2* fosters cell proliferation by promoting the degradation of *WEE1* mRNA, which negatively regulates cell-cycle-dependent kinases and proteins, including MYT1 [27]. The depletion of *YTHDF2* leads to an excess accumulation of WEE1 protein, causing a delay in the transition from the G2 to the M phase and hindering cell proliferation [27]. Additionally, previous research has indicated that YTHDF2 facilitates the mRNA degradation of specific target genes within cells [27]; however, the precise molecular mechanism by which *YTHDF2* regulates *PCNA and CDK4* expression in GCs remains unclear, representing a limitation of our study.

ERK1/2, as extracellular signal-regulated protein kinases, play a role in various physiological processes, including cell proliferation and apoptosis. Numerous studies have investigated the regulation of downstream gene expression via the ERK1/2 signalling pathway. BDNF has been shown to activate the ERK1/2 pathway, enhancing cell proliferation in bovine follicular granulosa cells [9] and stimulating proliferation in endometrial epithelial cells through TrkB-mediated ERK1/2 pathway activation [30]. In *rats*, BDNF has been found to inhibit GCs apoptosis via the ERK1/2 signalling pathway [31]. In our study, the results indicate that BDNF activates the ERK1/2 signalling pathway, leading to YTHDF2 upregulation in GCs. Conversely, blocking the ERK1/2 pathway significantly reduces YTHDF2 expression. This suggests that BDNF regulates YTHDF2 expression through the ERK1/2 pathway. Furthermore, in the context of Glioblastoma (GBM), EGFR/SRC/ERK signalling phosphorylates YTHDF2 at serine39 and threonine381, stabilizing the YTHDF2 protein. *YTHDF2* is essential for GBM cell proliferation, invasion, and tumourigenesis [32]. In summary, our findings indicate that *BDNF* modulates *YTHDF2* expression via the ERK1/2 signalling pathway.

## 4. Materials and Methods

### 4.1. Cell Culture and Treatments

This research received approval from the Ethics Committee of Jilin University and was conducted in compliance with animal welfare guidelines (SY202311100). Ovaries from *prepubertal Large White gilts* were obtained from a local slaughterhouse and transported to our laboratory in a 0.9% saline solution at 37 °C. The ovaries underwent sequential washings with a dilute bromogeramine solution, 70% alcohol for disinfection, and a 0.9% saline solution. Follicular fluid from 3–5 mm diameter follicles was collected using the aspiration method, then centrifuged at 600× *g* for 5 min to isolate GCs. These GCs were cultured in DMEM (High Glucose) medium (Bio-Channel, Nanjing, China) supplemented with 10% (*v*/*v*) foetal bovine serum (FBS) (Gibco, NY, USA) and maintained at 37 °C in a humidified atmosphere containing 5% CO_2_. When GC confluency reached 50–70%, the cells were treated with serum-free DMEM (High Glucose) containing brain-derived neurotrophic factor (BDNF) (Novoprotein, Suzhou, China) for 24 h. The cells were also exposed to 10 µM PD98059, a mitogen-activated protein kinase (MAPK-ERK1/2) activity inhibitor (Beyotime Biotechnology, Shanghai, China). After 30 min, this medium was replaced with fresh DMEM (High Glucose) containing 10% serum for an additional 24 h. Subsequently, the GCs were harvested for RT-qPCR and Western blot analyses.

### 4.2. Cellular Immunofluorescence

When the confluency of granulosa cells (GCs) in a 6-well plate reached 50–70%, the cells were washed three times for 5 min each time with PBS. Subsequently, the cells were fixed for 20 min in methanol at 4 °C and then washed three times for 5 min each time with PBS at 37 °C. The cells were blocked for 30 min at room temperature using 1% goat serum (*v*/*v*). They were then incubated overnight at 4 °C with a rabbit anti-FSHR polyclonal antibody (dilution 1:200; accession number: BS5724; Bioworld, Nanjing, China) or primary antibody diluent. Following this, the cells were washed three times with TBST for 5 min each time and incubated with a fluorescein isothiocyanate (FITC)-conjugated goat anti-rabbit secondary antibody (dilution 1:200; Bioworld, Nanjing, China) for 1 h at room temperature in the dark. The cells were then washed again three times with TBST for 5 min each. Next, the cells were incubated with DAPI solution/stain (dilution 1:1000; Beyotime, Nanjing, China) for 10 min in the dark. The stained GCs were examined using an Olympus fluorescence microscope (IX71; Olympus, Tokyo, Japan), and images were captured using the Olympus application suite (cellSens Dimension; Olympus, Tokyo, Japan).

### 4.3. Cell Viability Assay and Cell Counting Assay

The Cell Counting Kit-8 (CCK-8; Beyotime, Shanghai, China) was utilized to assess cell viability. GCs were seeded into 96-well plates. Once they reached 50–70% confluency, the cells were treated with varying concentrations of BDNF (0, 10, 40, 75, and 150 ng/mL). The siRNA targeting *YTHDF2* (siYTHDF2) combined with BDNF, PD98059, and BDNF in conjunction with PD98059 in the serum-free DMEM (High Glucose) medium for 24 h. After treatment, 10 μL of the CCK-8 solution was added to each well and incubated for 3–5 h, depending on cell density. Cell viability was then measured using an ELX 800 Universal Microplate Reader (BioTek, Highland Park, IL, VT, USA). These measurements were performed in triplicate at an absorbance wavelength of 450 nm (OD, 450 nm).

Cell proliferation was evaluated using a cell-counting assay. GCs were cultured in 24-well plates at a density of 1.5 × 10^5^ cells per well in 1 mL of medium. Following 24 h of treatment with BDNF, siYTHDF2, or siYTHDF2 + BDNF, the cells were washed, trypsinised, and subsequently counted using a Countess^TM^3 Automated Cell Counter (Invitrogen, Singapore).

### 4.4. Flow Cytometry

After 24 h of treatment with BDNF, siYTHDF2, siYTHDF2 + BDNF, PD98059, or BDNF + PD98059, GCs were harvested and placed into 1.5 mL centrifuge tubes. The cells were fixed with 70% ethanol at 4 °C for 12 h. Subsequently, the GCs were stained as per the protocol provided by the Cell Cycle and Apoptosis Analysis Kit (Beyotime, Shanghai, China). Cell cycle kinetics were analysed using a BD LSR flow cytometer (BD Biosciences, Franklin Lakes, NJ, USA).

### 4.5. Transfection with Small Interfering RNA

siRNAs were designed and synthesized by Suzhou GenePharma Co., Ltd. siYTHDF2 and a non-targeting siRNA serving as a negative control were transfected at a concentration of 50 nM (GenePharma, Suzhou, China). The sequences of siYTHDF2 were 5′-GGGCUGAUAUUGCUAGCAATT-3′ and 5′-UUGCUAGCAAUAUCAGCCCTT-3′. The siRNAs were mixed with Lipofectamine 3000 (Invitrogen, Carlsbad, CA, USA) and incubated in Opti-MEM Reduced Serum Medium (Gibco BRL, Gaithersburg, MD, USA) for 15 min. This mixture was then added to GCs cultured in serum-free DMEM (High Glucose) medium and incubated for 5–6 h. Afterward, the medium was replaced with fresh DMEM (High Glucose) medium containing 10% FBS or serum-free DMEM (High Glucose) medium supplemented with BDNF for an additional 24 h. The cells were subsequently harvested for RT-qPCR and Western blot analyses.

### 4.6. RNA Extraction, RT-PCR, and RT-qPCR Analysis

The total RNA from GCs was extracted using TRIzol reagent (Invitrogen, Foster City, CA, USA) following the manufacturer’s protocol. cDNA was synthesized using a Prime Script RT Reagent Kit with a gDNA Removal Kit (Takara Bio, Otsu, Japan). The total volume for the RT-qPCR reaction was 20 μL, comprising 10 μL of SYBR Green (Takara Bio, Otsu, Japan), 0.5 μL of each forward and reverse primer, 4 μL of cDNA, and 5 μL of DEPC-treated water. The RT-qPCR cycling conditions were as follows: initial denaturation at 95 °C for 10 min, followed by 40 cycles of denaturation at 95 °C for 30 s, and annealing at 60 °C for 1 min. The data were analysed using the comparative 2^−ΔΔCt^ method, with normalization to the GAPDH housekeeping gene. Primers were custom synthesized by Comate Bioscience Co., Ltd. (Changchun, China) (Table 1).

### 4.7. Western Blot Analysis

For protein isolation, cells were lysed using RIPA lysis buffer (Beyotime Biotechnology, Shanghai, China) supplemented with 1 mM PMSF (Beyotime Biotechnology, Shanghai, China). The extracted total proteins were then separated using 12.5% SDS-PAGE and subsequently transferred to a polyvinylidene difluoride membrane (0.2 μm; Millipore, Burlington, MA, USA) utilizing a semi-dry transfer system (Bio-Rad; Hercules, CA, USA). The membrane was blocked with a rapid blocking solution (YaMei, Hangzhou, China) for 12 min. After blocking, the membranes were incubated overnight with primary antibodies at 4 °C. Subsequently, the membranes were washed four times with TBST, each wash lasting 9 min. The membranes were then incubated with appropriate secondary antibodies for 1 h at room temperature, followed by four additional 9 min washes with TBST. For detection, the membrane was incubated with Enhanced Chemiluminescence (ECL) detection reagent (Thermo Scientific, Rockford, IL, USA) and imaged using a chemiluminescence detector (Tanon, Shanghai, China). Protein bands were scanned in grayscale using a Tanon Gel Imaging System (Tanon, Shanghai, China). Protein levels were quantified using ImageJ software (1.8.0), with each protein normalized against vinculin levels. The specific antibodies utilized are detailed in Table 2. 

### 4.8. Sequencing and Data Processing

Total RNA was extracted from GCs, and assessments of RNA purity and quality were conducted. The construction of the cDNA library and a subsequent sequencing analysis were carried out in collaboration with Hangzhou Lianchuan Biotechnology Co., Ltd. (Hangzhou, China).

### 4.9. Statistical Analysis

All statistical analyses were conducted using SPSS version 19.0 (SPSS Corp., Armonk, NY, USA). Each experiment was repeated a minimum of three times with at least three independent replicates. To evaluate statistical differences between two groups, an independent samples *t*-test was utilized, whereas differences among more than two groups were assessed using one-way analysis of variance (ANOVA). The data are presented as the mean ± standard error of the mean (SEM). *p* < 0.05 (*) was considered significant.

## 5. Conclusions

This study is the first to demonstrate that *BDNF* upregulates the expression of *YTHDF2* by activating the ERK1/2 pathway. In turn, *YTHDF2* enhances the expression of *CDK4 and PCNA*, thereby promoting the viability and proliferation of GCs (Figure 6). This research provides a novel perspective and foundational basis for further exploring the mechanisms through which BDNF regulates the viability and proliferation of GCs. This opens up new possibilities for developing biomedical procedures for the BDNF-triggered stimulation of GC-based regenerative and reconstructive medicine in female patients exhibiting ovarian dysfunctions.

## Figures and Tables

**Figure 1 ijms-25-02343-f001:**
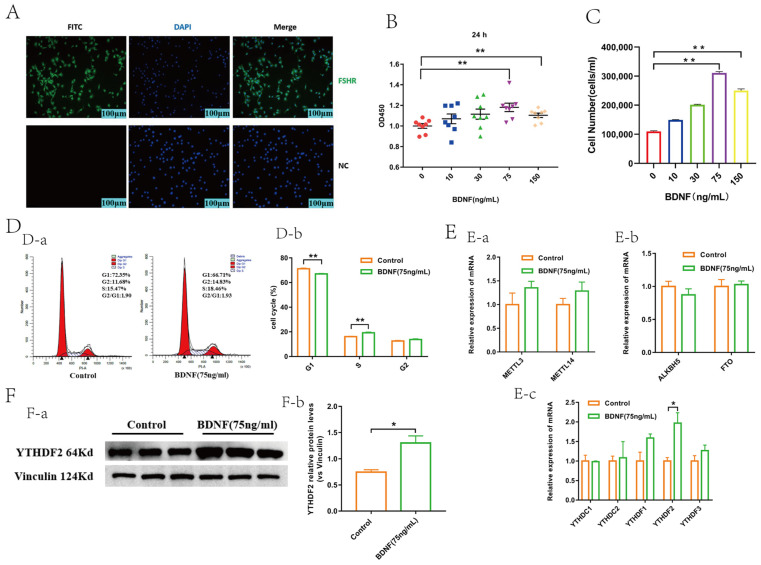
BDNF promoted the proliferation and upregulated expression of YTHDF2 in GCs. (**A**) The characterization of GCs by cell immunofluorescence. The GCs were incubated with rabbit anti-FSHR polyclonal antibody or primary antibody dilution (NC, negative control) and then coupled with FITC-conjugated secondary antibody (FITC; green) and diamidino phenylindole (DAPI; blue) for staining cell nuclei. The image (merge) was the result of the merging of green and blue. Scale bar = 100 μm. FSHR: follicle stimulating hormone receptor. (**B**) The viability of GCs treated with various concentrations of BDNF at 24 h determined by CCK-8 assay (*n* = 8). (**C**) GC number were determined by cell-counting assay (*n* = 3). (**D-a**) The change in the cell cycle in GCs determined by flow cytometry (*n* = 3). (**D-b**) Statistics of the proportion of each cell cycle (*n* = 3). (**E-a**) The effect of BDNF on m^6^A methylase mRNA level expression in GCs (*n* = 3). (**E-b**) The effect of BDNF on m^6^A demethylase mRNA level expression in GCs (*n* = 3). (**E-c**) BDNF promoted the expression of YTHDF2 in mRNA level (*n* = 3). (**F-a**) BDNF promoted the expression of YTHDF2 in protein level (*n* = 3). (**F-b**) Statistical analysis of gray value of protein bands (*n* = 3). To evaluate statistical differences between two groups, an independent samples *t*-test was utilized, whereas differences among more than two groups were assessed using one-way analysis of variance (ANOVA). All values are presented as the mean ± SEM. All significant differences are from the treatment group compared to the control group (*, *p* < 0.05; **, *p* < 0.01).

**Figure 2 ijms-25-02343-f002:**
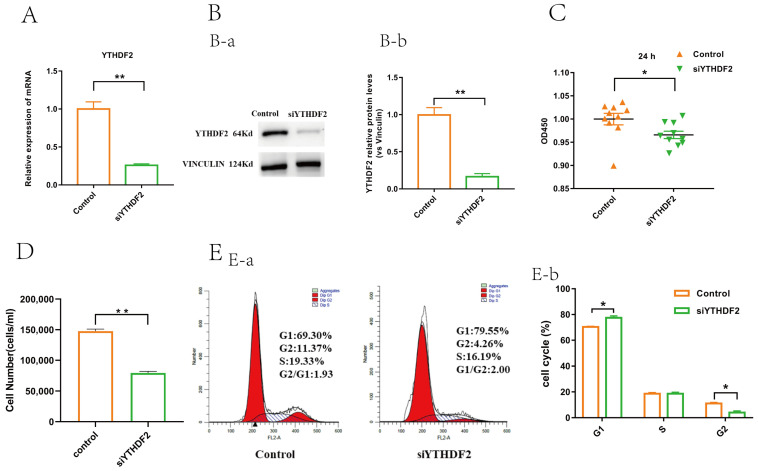
YTHDF2 knockdown inhibited proliferation of GCs. (**A**) mRNA level of YTHDF2 downregulated after treatment with siYTHDF2 determined by RT-qPCR (*n* = 3). (**B-a**) Protein level of YTHDF2 downregulated after treatment with siYTHDF2 determined by WB (*n* = 3). (**B-b**) Statistical analysis of gray value of protein bands (*n* = 3). (**C**) Viability of GCs was inhibited by CCK-8 (*n* = 10). (**D**) GC number were determined by cell-counting assay (*n* = 3). (**E-a**) Comparison of cell cycle between control group and siYTHDF2 group determined using flow cytometry (*n* = 3). (**E-b**) Statistics of the proportion of each cell cycle between control group and siYTHDF2 group (*n* = 3). All values are presented as mean ± SEM. All experimental analyses were conducted through independent samples *t*-test. All significant differences are from siYTHDF2 group compared to control group (*, *p* < 0.05; **, *p* < 0.01).

**Figure 3 ijms-25-02343-f003:**
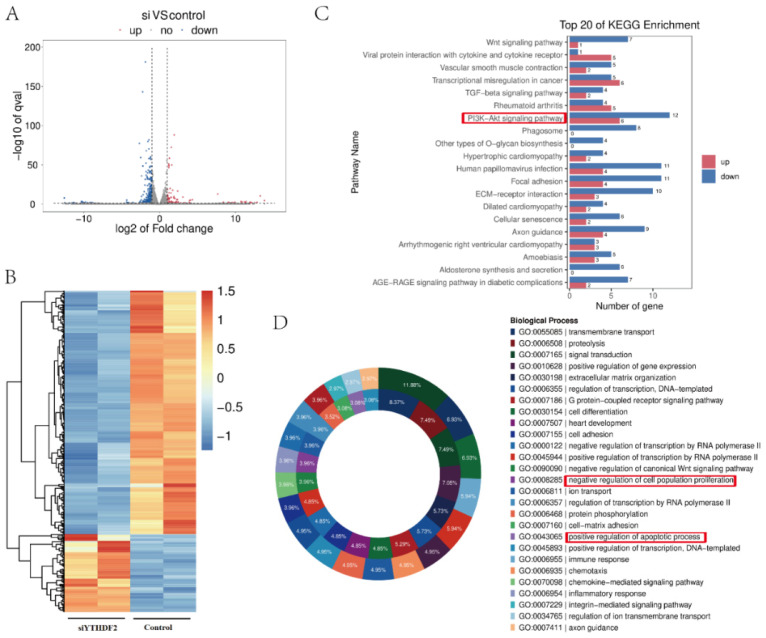
Transcriptome sequencing. (**A**) The volcanic map of differential genes between the SiYTHDF2 and control group. (**B**) The heatmap of differential genes, in which the horizontal coordinate of the heat map is the sample, and the vertical coordinate is the screened differentially expressed genes. (**C**) The top 20 pathways with the smallest *p* value were screened using a KEGG enrichment analysis. (**D**) The GO entries with the most significant genes are shown using the doughnut chart. Sample of transcriptome sequencing was 2 in each group. The experiment of transcriptome sequencing was performed once. All results are from the siYTHDF2 group compared to the control group. The difference threshold is set to (|log2FC|> = 1 & *p* < 0.05).

**Figure 4 ijms-25-02343-f004:**
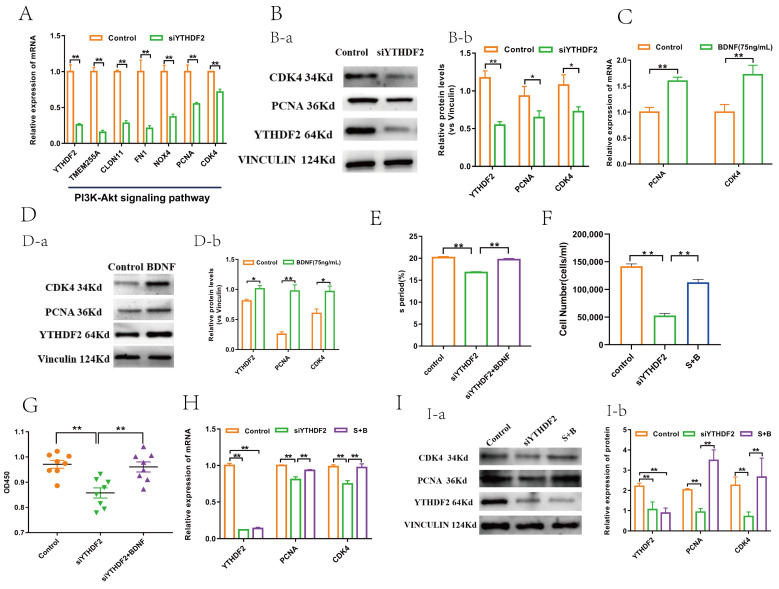
YTHDF2 promotes GC proliferation by upregulating CDK4 and PCNA. (**A**) RT-qPCR was used to verify mRNA level in YTHDF2 and top six genes with significant differences in PI3K-Akt signalling pathway from sequencing data (*n* = 3). (**B-a**) Protein level change in YTHDF2, PCNA, and CDK4 after YTHDF2 knockdown (*n* = 3). (**B-b**) Statistical analysis of gray value of protein bands (*n* = 3). (**C**) mRNA level change in CDK4 and PCNA after BDNF treatment (*n* = 3). (**D-a**) Protein level change in YTHDF2, CDK4, and PCNA after BDNF treatment (*n* = 3). (**D-b**) Statistical analysis of gray value of protein bands (*n* = 3). (**E**) Comparison of S period among three groups determined by flow cytometry (*n* = 3). (**F**) Comparison of GC number determined by cell-counting assay (*n* = 3). (**G**) Comparison of viability of GCs (*n* = 8). (**H**) mRNA level change in YTHDF2, CDK4, and PCNA (*n* = 3). (**I-a**) Protein level change in YTHDF2, CDK4, and PCNA (*n* = 3). (**I-b**) Statistical analysis of gray value of protein bands (*n* = 3). All values are presented as mean ± SEM. To evaluate statistical differences between two groups, independent samples *t*-test was utilized, whereas differences among more than two groups were assessed using one-way analysis of variance (ANOVA). (*, *p* < 0.05; **, *p* < 0.01). S + B = siYTHDF2 + BDNF.

**Figure 5 ijms-25-02343-f005:**
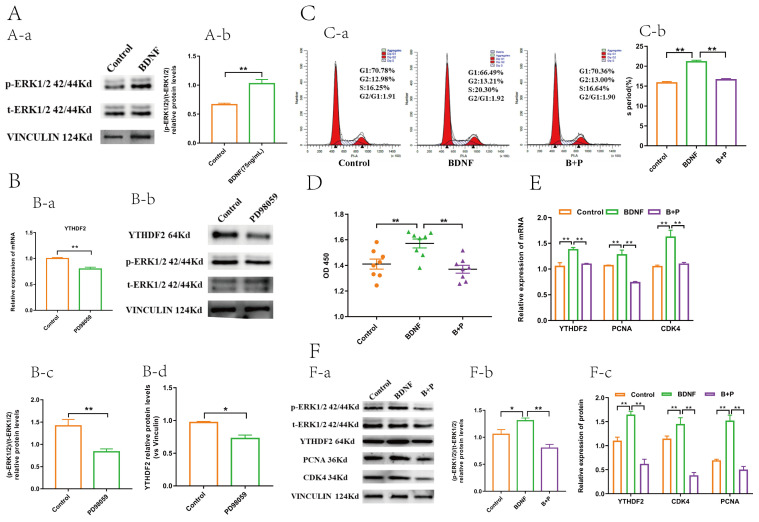
BDNF upregulated YTHDF2 through increasing the phosphorylation level of the ERK1/2 signal pathway. (**A-a**) Protein level change in t-ERK1/2 and p-ERK1/2. (*n* = 3). (**A-b**) Statistical analysis of the ratio of p-ERK1/2 to t-ERK1/2 of gray value of protein bands (*n* = 3). (**B-a**) The mRNA level change in YTHDF2 after PD98059 treatment (*n* = 3). (**B-b**) the protein level change in YTHDF2, ERK1/2, and p-ERK1/2 after PD98059 treatment (*n* = 3). (**B-c**) Statistical analysis of the ratio of p-ERK1/2 to t-ERK1/2 of gray value of protein bands after PD98059 treatment (*n* = 3). (**B-d**) Statistical analysis of gray value of YTHDF2 protein bands after PD98059 treatment (*n* = 3). (**C-a**) Comparison of the S period among the three groups determined by flow cytometry (*n* = 3). (**C-b**) Statistics of the proportion of S period among the three groups (*n* = 3). (**D**) Comparison of GC viability among the three groups determined by flow cytometry (*n* = 8). (**E**) mRNA level change in YTHDF2, PCNA, and CDK4 among the three groups (*n* = 3). (**F-a**) Protein level change in t-ERK1/2, p-ERK1/2, YTHDF2, PCNA, and CDK4 among the three groups (*n* = 3). (**F-b**) Statistical analysis of the ratio of p-ERK1/2 to t-ERK1/2 of gray value of protein bands among the three groups (*n* = 3). (**F-c**) Statistical analysis of gray value of YTHDF2, CDK4 and PCNA protein bands among the three groups (*n* = 3). All values are presented as the mean ± SEM. To evaluate statistical differences between two groups, an independent samples *t*-test was utilized, whereas differences among more than two groups were assessed using one-way analysis of variance (ANOVA). (*, *p* < 0.05; **, *p* < 0.01). B + P = BDNF + PD98059.

**Figure 6 ijms-25-02343-f006:**
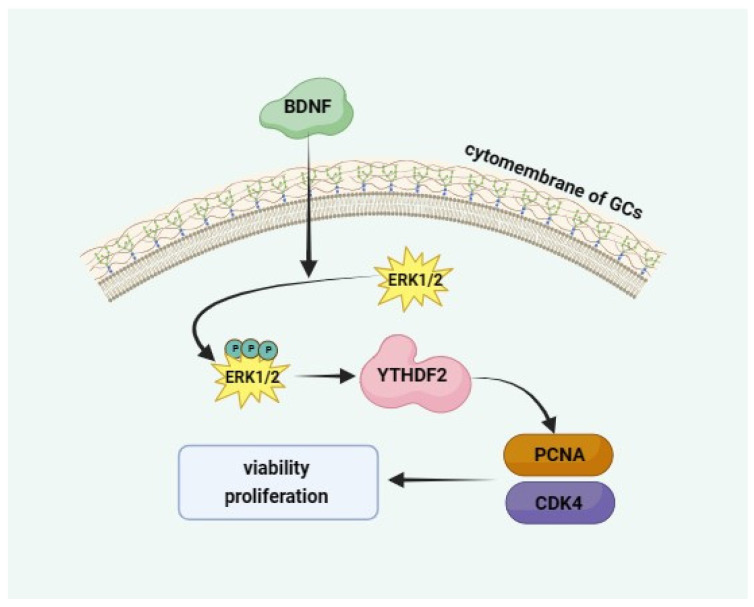
A schematic illustration of YTHDF2 mediating the action of BDNF stimulating the proliferation of porcine follicle granulosa cells. BDNF activates the ERK1/2 pathway to upregulate YTHDF2 expression, and YTHDF2 upregulates CDK4 and PCNA expression to promote GC proliferation. Black arrows represent expression promotion.

**Table 1 ijms-25-02343-t001:** The sequences of primers used in RT-qPCR.

Name	Sequences	Products Sizes, bp	Melting Temperature, °C
*GAPDH*	F-5′ TCGGAGTGAACGGATTTGGC3′	147	83.6
	R-5′ TGCCGTGGGTGGAATCATAC3′		
*YTHDF2*	F-5′ CAGGCAAGGCCCAATAATGC3′	186	86.1
	R-5′ TCTCCGTTGCTCAGTTGTCC3′		
*CDK4*	F-5′ ATGGGACCGTGTACAAAGCA3′	165	81.2
	R-5′ CATCCATCAGCCGGACAACA3′		
*PCNA*	F-5′ ATGCAGACACCTTGGCACTA3′	153	79.9
	R-5′ ACGTGCAAATTCACCAGAAGG3′		
*TMEM255A*	F-5′ TCCTCAAGTGGCCTCCTACA3′	190	88.2
	R-5′ TGGAGAGTATCGGGGTGGAG3′		
*CLDN11*	F-5′ GTGACCTGCGGCTACACTAT3′	182	91.1
	R-5′ TCGGCAAGCCTGAACATAGC3′		
*FN1*	F-5′ GCACCATCCAACTTGCGTTT3′	179	88.1
	R-5′ TGTACTCGGTTGCTGGTTCC3′		
*NOX4*	F-5′ TCCTGGCTTACCTTCGAGGA3′	146	85.4
	R-5′ TTCACGGAGAAGTTGAGGGC3′		

**Table 2 ijms-25-02343-t002:** Antibody information and dilutions in this study.

Antibodies Name	Diluted Multiples	Accession Number	Reagent Company
Rabbit anti-vinculin polyclonal antibody	1:750	BS62273	Bioworld
Rabbit anti-YTHDF2 monoclonal antibody	1:750	ab220163	Abcam
Rabbit anti-CDK4 polyclonal antibody	1:750	ab95255	Abcam
Mouse anti-PCNA monoclonal antibody	1:750	MB66871	Bioworld
Rabbit anti-ERK1/2 polyclonal antibody	1:750	BS90472	Bioworld
Rabbit anti-phospho-ERK1/2 polyclonal antibody	1:750	BS4621P	Bioworld
Goat anti-rabbit IgG antibody	1:8000	BS13278	Bioworld
Goat anti-mouse IgG antibody	1:8000	BS12478	Bioworld

## Data Availability

All data involved in this article are original and available from the corresponding authors upon reasonable request.

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
