# Peer review of "YTHDF2 as a Mediator in BDNF-Induced Proliferation of Porcine Follicular Granulosa Cells"

_ijms, 2024, doi:10.3390/ijms25042343_

Round 1

Reviewer 1 Report (New Reviewer)

Comments and Suggestions for Authors

The current study is a very well written article providing excellent interpretation of mechanistic interrelations at the transcriptomic and proteomic levels, which exert the significant impacts on the proliferation of porcine GCs, forming a basic physiological and histological compartment in ovarian follicles. To the best of my knowledge, this work is the first report that has proved the involvement of m6A reader designated as YTHDF2 in the BDNF-prompted proliferation of porcine GCs within the framework of the completely new in vitro models of follicular cell culture engineering. This opens up the new possibilities for developing biomedical procedures of BDNF-triggered stimulation of GC-based regenerative and reconstructive medicine in female patients exhibiting ovarian dysfunctions.

The methodology, statistical analyses and the results are comprehensively described, which confirms the selection of relevant procedures by the Authors and the multifaceted interpretation of the results achieved.

To sum up, the manuscript has to be only slightly revised according to minor points shown below:

1) Please add (at the very end of Conclusions section) the possible practical implications of the results obtained by the Authors as has been above-indicated by the Reviewer: "This opens up the new possibilities for developing biomedical procedures of BDNF-triggered stimulation of GC-based regenerative and reconstructive medicine in female patients exhibiting ovarian dysfunctions."

2) The Abbreviations section, in which all the in-text abbreviations should be diligently expanded, has to be added at the end of the article (below the Conclusions section).

3) The References have to be prepared in the format adequate to IJMS requirements.

Author Response

Reviewer 2 Report (New Reviewer)

Comments and Suggestions for Authors

The paper under review provides an in-depth investigation into the mechanism of the mitogenic effect of Brain-Derived Neurotrophic Factor (BDNF) on mammalian granulosa cells. Given the significance of this topic in the realms of reproductive biology and potential implications for preserving female fertility in both animal husbandry and human medicine, the research is particularly timely and pertinent.

The methodological rigor and experimental quality employed in this study are commendable. The adequacy of the experimental methods and the high-quality execution of these procedures significantly enhance the credibility and scientific rigor of the findings. The experimental results, as illustrated in an acceptable manner, contribute to the overall clarity and coherence of the research.

The reliability of the data obtained from this study is robust and indubitable. The conclusions drawn from the analysis and interpretation of the data are well-founded, logical, and consistent with the objectives of the study.

Based on the methodological soundness, the quality of the experimental work, and the reliability of the findings, I wholeheartedly recommend the publication of this article in its current form. The depth of insight and the significance of the study's findings make it a valuable addition to the current literature and a potential catalyst for further advancements in the understanding of the role of BDNF on regulation of mammalian granulosa cells state.

Author Response

This manuscript is a resubmission of an earlier submission. The following is a list of the peer review reports and author responses from that submission.

Round 1

Reviewer 1 Report

Comments and Suggestions for Authors

The article by Liu et al., described the role of YTHDF2 as a downstream mediator of BDNF signaling during porcine granulosa cell (GC) proliferation. While the role of BDNF in regulating GC proliferation is well studied in several mammalian model system, this report shows a role of m6A reader YTHDF2 for the first time. The research is well conducted and the conclusion is supported by the experiments. I have the following comments:

1. What is the kinetics of the ERK activation due to BDNF treatment? Does that temporally match with YTHDF2  expression?

2. What is the purpose of Fig. 3?

The writing within the graphs are not readable (D). 

While the authors found several signaling pathway by the KEGG enrichment (which is expected), what is the point of showing them all, since majority of them were not used/described by the authors. e.g., what is amoebiasis pathway and why that is related?  

3. How does the histograms for WB were generated?

Reviewer 2 Report

Comments and Suggestions for Authors

Abstract

1. I think that it size can be reduced

2. What the abbreviation GO does mean?

Methods

1. Why did you use only one gene as a reference? Why did you selected particularly GAPDH as a reference gene?

2. lines 356-357 “YTHDF2(1:750, abcom) CDK4(1:750, abcom)” Abcom? Maybe Abcam?

3. Statistical analysis. Why did you used parametric criteria for analysis? Describe it. How did you verified normal distribution of the data?

4. Add to the table 1 information about products sizes and melting temperature of primers.

Round 2

Reviewer 1 Report

Comments and Suggestions for Authors

The authors' answers are satisfactory. This reviewer is in agreement with publication.